# Differences in Regulatory Mechanisms Induced by β-Lactoglobulin and κ-Casein in Cow’s Milk Allergy Mouse Model–In Vivo and Ex Vivo Studies

**DOI:** 10.3390/nu13020349

**Published:** 2021-01-25

**Authors:** Dagmara Złotkowska, Emilia Stachurska, Ewa Fuc, Barbara Wróblewska, Anita Mikołajczyk, Ewa Wasilewska

**Affiliations:** 1Department of Immunology and Food Microbiology, Institute of Animal Reproduction and Food Research of the Polish Academy of Sciences, Tuwima 10 Str., 10-748 Olsztyn, Poland; zimzpan@gmail.com (E.S.); e.fuc@pan.olsztyn.pl (E.F.); b.wroblewska@pan.olsztyn.pl (B.W.); 2Department of Public Health, Faculty of Health Sciences, Collegium Medicum, University of Warmia and Mazury, 10-082 Olsztyn, Poland; anita.mikolajczyk@uwm.edu.pl

**Keywords:** β-lactoglobulin, κ-casein, cow’s milk hypersensitivity, T cells, mouse model

## Abstract

The presence of various proteins, including modified ones, in food which exhibit diverse immunogenic and sensitizing properties increases the difficulty of predicting host immune responses. Still, there is a lack of sufficiently reliable and comparable data and research models describing allergens in dietary matrices. The aim of the study was to estimate the immunomodulatory effects of β-lactoglobulin (β-lg) in comparison to those elicited by κ-casein (κ-CN), in vivo and ex vivo, using naïve splenocytes and a mouse sensitization model. Our results revealed that the humoral and cellular responses triggered by β-lg and κ-CN were of diverse magnitudes and showed different dynamics in the induction of control mechanisms. β-Lg turned out to be more immunogenic and induced a more dominant Th1 response than κ-CN, which triggered a significantly higher IgE response. For both proteins, CD4^+^ lymphocyte profiles correlated with CD4^+^CD25^+^ and CD4^+^CD25^+^Foxp3^+^ T cells induction and interleukin 10 secretion, but β-lg induced more CD4^+^CD25^+^Foxp3^-^ Tregs. Moreover, ex vivo studies showed the risk of interaction of immune responses to different milk proteins, which may exacerbate allergy, especially the one caused by β-lg. In conclusion, the applied model of in vivo and ex vivo exposure to β-lg and κ-CN showed significant differences in immunoreactivity of the tested proteins (κ-CN demonstrated stronger allergenic potential than β-lg), and may be useful for the estimation of allergenic potential of various food proteins, including those modified in technological processes.

## 1. Introduction

Cow’s milk is a rich source of nutritional compounds for neonates, but it is also a source of potential allergens. Milk consists of whey proteins and casein, whose susceptibility to proteolysis varies, and therefore their allergenic potential, so they can stimulate different mechanisms of the immune system. This is relevant where whey proteins are ubiquitous constituents of processed foods, used for desirable properties such as high nutritional and dietary value, antioxidant and antimicrobial properties, and the ability to act as foaming or emulsifying agents [1]. They are present in milk-based formulas for infants and in food products, including protein bars and drinks targeted at specific consumer groups, including professional athletes and bodybuilders [2].

β-Lactoglobulin (β-lg) is a major whey protein alongside α-lactalbumin (α-la), present in cow’s milk at a concentration of approximately 3–4 g/L [3]. It is found in the milk of most mammalian species, except for humans, rodents, and lagomorphs (rabbits and hares). For years, β-lg has attracted the attention of researchers as the main allergenic milk protein because it does not occur naturally in human milk. The β-lg molecule comprises 162 amino acid residues and has a molecular weight of 18.4 kDa, with a tertiary structure consisting of two anti-parallel β-sheets that form a binding pocket and connected by an α-helical strand. β-Lg predominantly exists as a dimer, but can also form other quaternary structures [3,4]. The ligand-binding pocket of β-lg is a structural feature that links it to the lipocalin family, whose functions encompass binding and transport of hydrophobic molecules, such as fatty acids, retinoids, and steroids [5]. In addition to caseins, β-lg is regarded as a major milk allergen that can induce adverse immune responses in sensitive individuals [6]. Both IgE and IgG epitopes were identified in the β-lg structure and have been proven to react with cow’s milk allergy (CMA) patient serum samples [7].

κ-Casein (κ-CN) is a type of casein protein that acts as one of the primary milk’s allergens. It has a molecular weight of approximately 19 kDa, close to β-lg, and is present in milk in concentrations like that of β-lg [3]. κ-Casein facilitates the formation of casein micelles by acting as an interface between the hydrophobic core of the casein micelles and the aqueous environment. Among the four fractions of casein, κ-CN seems to exhibit the smallest allergenic potential. However, it has IgE-binding regions, find in CMA patients over 7 years old [8].

Food proteins modulate the mucosal immune system response via inductive sites such as Payer’s patches (PPs) and mesenteric lymph nodes (MLNs) [9,10]. They may affect the humoral responses inducing IgG, IgA, or IgE production and, depending on food matrix, the level of immunocompetent cells controlling the balance between allergy and immunity, and consequently intestinal homeostasis [11,12,13]. Intestinal CD4^+^ lymphocytes play a pivotal role in the induction of the immune response against food antigens. Upon local activation by antigen presenting cells, naïve CD4^+^ lymphocytes expand rapidly and acquire one of several phenotypes, which are characterized by the expression of cellular and intracellular markers and specific secretory activity, e.g., regulatory cytokines, and subsequently differentiate into Th1 and Th2 subpopulations [14]. The activity of induced subsets of CD4^+^ T lymphocytes can lead to system-wide effects, such inflammation and allergy. In some cases, Th2 lymphocytes may contribute to local inflammation of the intestines [15]. Additionally, T helper 17 (Th17) cells can exert proinflammatory effects by secreting interleukin 17 (IL-17) and can act independently or in concert with Th1 and Th2 cells [16]. Of the activated CD4^+^ cells, the exception is T CD4^+^ regulatory lymphocytes (Tregs, regulatory T cells), which inhibit the immune response and are one of the key mechanisms to prevent autoimmunity, including response to food allergens [17]. Tregs are a heterogenous subpopulation of T lymphocytes which can inhibit the function of these cells which, in an immune response, perform executive functions, i.e., effector lymphocytes. The flexibility of effector CD4^+^ T lymphocytes and context-dependent acquisition of distinct phenotypes increases the complexity of intestinal immunity. Research is needed to get knowledge about the role of the gut associated CD4^+^ T lymphocytes with a memory phenotype in persistent immune reactions, such as food allergy or oral tolerance. Previously, we found differences in immunomodulatory properties of α- and κ-casein at the cellular level, indicating a stronger allergenicity of α-casein [18]. The present study aimed to investigate and compare the effect of two similar, in terms of source and molecular weight, milk proteins, the major whey allergen β-lg and presented rather as a weaker allergen κ-CN, on the immune response, with emphasis on regulatory mechanisms of the gut-associated lymphoid tissue. The humoral and cellular immune responses were characterized, and the activity of CD4^+^ T lymphocytes, including Tregs induction and cytokine secretion following stimulation with different milk antigens in vivo and ex vivo, have been selected as markers differentiating immunogenic activity of those two proteins.

## 2. Materials and Methods

### 2.1. Antigens

The proteins α-la, β-lg, α-casein (α-CN), and κ-CN used in this study were purchased from Sigma-Aldrich (Sigma-Aldrich, St. Louis, MO, USA). All antigens were of analytical grade.

### 2.2. Mice and Treatment Groups

Six-weeks-old female BALB/ccmdb mice were purchased from the Center of Experimental Medicine of the Medical University of Bialystok (Bialystok, Poland) and housed in the Animal Facility of the Institute of Animal Reproduction and Food Research of the Polish Academy of Sciences, Olsztyn, using an individually ventilated cage system (IVC; M.AC.S.^®^, Alternative Design manufacturing and Supply INC., USA). A total pathogen-free diet (1324 TPF, Altromin, Germany) devoid of cow’s milk proteins and sterile drinking water was provided ad libitum. All experimental procedures were approved by the Local Ethical Committee for Animal Experiments in Olsztyn (43/2015).

Groups of mice (8 mice per group) were immunized intraperitoneally (i.p.) with 100 µg (per mouse) of β-lg (group β-lg) or κ-CN (group κ-CN) in 100 µL of PBS with complete Freund’s adjuvant (1:1, vol/vol) and subsequently boosted after 7 and 14 days in the presence of incomplete Freund’s adjuvant (Figure 1A). Blood and fecal samples were collected starting from day 14 for determination of total IgE and specific IgG and IgA levels. From day 52 of the experiment, over the next 16 days, 200 µg of antigen was administered daily to each mouse via oral gavage. Mice were co-administered with 2.5 µg of cholera toxin (CT) (Sigma) as mucosal adjuvant on days 52, 59, and 66 [19]. As a control (group PBS), mice were treated with 200 μL of sterile PBS via i.p. and oral route. On day 68, the mice were subjected to euthanasia via CO_2_ inhalation (the CO_2_ flow rate displaced 10% of the chamber volume per minute) [18]. Spleen tissue (SPL), head and neck lymph nodes (HNLNs), mesenteric lymph nodes (MLNs), Peyer’s Patches (PPs), and blood were collected from each group for lymphocyte isolation. Non-immunized mice were used for lymphocytes isolation for the ex vivo proliferation index (PI) assay (Figure 1B).

### 2.3. Lymphocyte Isolation

SPL, MLNs, and HNLNs were homogenized in a glass homogenizer in 3 mL of incomplete medium (IM: RPMI−1640 containing L glutamine supplemented with 1 mM HEPES and 10 U/mL penicillin-streptomycin solution). Next, the cell suspension was filtered through an 80-µm mesh membrane and centrifuged at 400× *g* at 10 °C for 10 min (Eppendorf 5804 R, Hamburg, Germany). Cell pellets were suspended in 1 mL of IM for cell counting using the trypan blue exclusion method. Splenocytes were additionally treated with red blood cell buffer (Cat. No. 11814389001, Roche Diagnostics GmbH, Mannheim, Germany) for 5 min to remove the remaining red blood cells, then were washed and suspended in 1 mL of IM for cell counting using the trypan blue exclusion method.

### 2.4. Peripheral Blood Mononuclear Cell (PBMC) Isolation

Blood samples from each experimental group were mixed with 20 µL of Heparinum WZF (5000 IU/mL, Polfa, Warsaw, Poland) to prevent clotting. Each sample was mixed with an equal volume of PBS and separated via density gradient centrifugation using Histopaque^®^-1077 (Sigma-Aldrich). The interface layer containing the mononuclear cells was isolated and washed in IM, after which the pellet was resuspended in 1 mL of IM for counting.

### 2.5. Lymphocyte Proliferation Assay

Lymphocyte potential for antigen recognition was checked by determination of their proliferation activity ex vivo using MTT assay or flow cytometry with CFSE staining (Figure 1B).

#### 2.5.1. MTT Assay

For the MTT proliferation assay, splenocytes from naïve mice were seeded on 96-well culture Nunc™ plates at the density of 10^6^/mL in complete medium (CM: RPMI−1640 containing L glutamine and supplemented with 1 mM HEPES, 10 U/mL penicillin-streptomycin, 1 mM sodium pyruvate, 1 mM non-essential amino acids, and 10% heat-inactivated fetal bovine serum (FBS)) and incubated at 37 °C, 5% CO_2_, and 90% air humidity. After 12 h of culture stabilization, cells were stimulated with 10, 50, 100, and 200 µg/mL β-lg or κ-CN. Positive control cells were incubated with 10 µg/mL Concanavalin A (Con-A). MTT assay was performed according to the manufacturer’s instructions (Cat. No. 10009365; Cayman Chemical, Ann Arbor, MI, USA). After 120 h of culture, 10 µL of MTT reagent was added to each well. Next, cells were incubated for 4 h, and subsequently, 100 µL of crystal dissolving solution was added. Plates were incubated for another 4 h, and the absorbance was measured at λ = 570 nm using a UVM 340 spectrophotometer (ASYS-Hitech GmbH, Eugendorf, Austria). The lymphocyte proliferation index (PI) was calculated by dividing the absorbance of stimulated cells by the absorbance of unstimulated cells and expressed as a mean of the group (*n* = 6) ± SD.

#### 2.5.2. Carboxyfluorescein Succinimidyl Ester (CFSE) Assay

For the CFSE assay, cells isolated from SPL of control (PBS group) or experimental mice (β-lg and κ-CN groups) were suspended in 1 mL of IM containing 5% FCS, and subsequently 1.1 µL of 5.5 mM carboxyfluorescein succinimidyl ester (CFSE) dissolved in 110 µL of PBS was added. Cells were incubated for 5 min in the dark at RT, washed twice with PBS containing 5% FCS, and washed once with PBS containing 1% FCS. Cells were then centrifuged at 400× *g* at 10 °C for 10 min. The pellet was suspended with appropriate amounts of CM and plated on 96-well plates at 10^6^ cells/mL. Cells were then allowed to recover by incubation for 12 h at 37 °C, 5% CO_2_, and 90% air humidity. On the next day, cells were stimulated with 200 µg/mL antigen or 10 µg/mL Con-A as a positive control. After 120 h of incubation, cells were collected, washed, and centrifuged twice (400× *g*; 10 °C; 10 min) with FACS buffer (1% FCS in PBS). Afterwards, cells were fixed with 250 µL of 2% paraformaldehyde (PFA). Cytometric assay was performed using BD LSRFortessa Cell Analyzer (BD Bioscience, San Jose, CA, USA) equipped with DIVA software (BD Bioscience). Results were analyzed using FlowJo^®^ (BD Bioscience) statistical modelling software. The lymphocyte PI was calculated as the average number of divisions excluding undivided cells. Samples were analyzed in triplicates and results are presented as the mean of PI in the group ± SD.

### 2.6. Lymphocyte Phenotyping

Lymphocytes were stained for cellular markers using an antibody cocktail containing the following antibodies: PE anti-mouse CD3, AF700 anti-mouse CD8α, FITC anti-mouse CD4, and PerCP-Cy5.5 anti-mouse CD25. After fixation and permeabilization, cells were intracellular stained with APC anti-mouse Foxp3. Next, cells were washed and fixed in 250 µL of 2% PFA and analyzed using a BD LSRFortessa cell analyzer (BD Biosciences, San Jose, CA, USA). All reagents used for cytometric analysis were purchased from BD Biosciences (San Jose, CA, USA).

### 2.7. Serum Samples

Blood samples obtained via submandibular vein puncture were incubated at 20 °C (RT) for 30 min and centrifuged at 16,900× *g* using an Eppendorf 5418R centrifuge (Eppendorf, Hamburg, Germany). Serum samples were collected and stored at −20 °C for subsequent analysis [20].

### 2.8. Fecal Extracts

Collected fecal samples were extracted in PBS (0.1 M phosphate buffered saline, pH 7.2, containing 0.1% NaN_3_), with a weight-to-volume ratio of 1:10. Samples were homogenized in a mechanical shaker for 10 min at 4 °C using Fugamix^®^ (ELMI, ILD, Latvia) and then centrifuged at 16,900× *g* for 10 min at 10 °C. The supernatants were stored at −20 °C for further analysis [20].

### 2.9. ELISA Assay of Total and Specific Antibodies

Indirect ELISA was used for specific antibodies assay. A 96-well Microlon 600 microplate (Greiner Bio-One Gmbh, Frickenhausen, Germany) was coated with antigen dissolved in PBS at a concentration of 20 µg/mL and incubated at 4 °C overnight. On the following day, wells were blocked with 150 µL of 1% BSA in PBS, incubated for 1 h at 37 °C, and subsequently washed four times with PBS-T (0.5% Tween 20 in PBS). Next, 50 µL of serial sample dilution was loaded onto each well. The plate was incubated overnight at 4 °C and washed four times with PBS-T. Then, 50 µL of either anti-mouse IgA or IgG peroxidase-conjugated antibody (Sigma-Aldrich) was added to the wells, after which the plate was incubated at 37 °C for 1 h and washed with PBS. Next, 50 µL of peroxidase substrate ABTS (Millipore, Temecula, CA, USA) was added to develop color reaction. Following 1 h of incubation at RT, the absorbance was measured at λ = 405 nm using a Jupiter UVM-340 spectrophotometer (ASSYS-Hitech GmbH, Eugendorf, Austria). The endpoint titer (EpT) was expressed as the reciprocal dilution of the last sample dilution with an absorbance of 0.1 OD units above that of the negative control [18].

Total IgE in serum was determined by a commercial Mouse IgE ELISA Set (Cat. No. 157718, Cambridge, MA, USA). Assay was performed according to the manufacture’s recommendations.

### 2.10. Cytokine Profiles

The levels of IL-6, IL-10, IL-17A, IFN-γ, and TNF-α secreted into the culture medium were assayed using a BD Mouse Th1/Th2/Th17 CBA Kit (BD Biosciences) according to the manufacturer’s instructions. Briefly, 50 µL of capture bead suspension, 50 µL of test sample or standard, and 50 µL of detection reagent were mixed in a separate tube and incubated at RT for 2 h. Samples were then washed with 1 mL of wash buffer and centrifuged at 400× *g* for 5 min. The supernatant was discarded, and the pellet was suspended in 300 µL of wash buffer and assayed on a BD LSRFortessa cell analyzer. Cytokine concentrations were calculated using FCAP Array™ 3.0 software (BD Bioscience).

### 2.11. Statistical Analysis

Data were analyzed with Prism 9 (GraphPad Software, LLC, San Diego, CA, USA). The values for all measurements were expressed as means ± SD. The *t* test was used to evaluate differences between two groups, whereas one-way ANOVA followed by post-hoc Tukey’s test were applied for multiple comparisons. The differences were considered statistically significant at *p* ≤ 0.05.

## 3. Results and Discussion

### 3.1. Preliminary Estimation of β-Lg and κ-CN Immunogenicity Based on the Antigen Dose

Different food proteins can induce varying magnitudes of immune responses, and the most reliable methods for determining the allergenicity of food protein are still under debate [21]. Previously, we proposed that the proliferative index (PI) of naïve lymphocytes determined ex vivo to be used as an initial screening for protein immunogenicity [18]. Now, we incubated naïve mouse lymphocytes in presence of varying antigen concentrations. Figure 2 shows the response of naïve lymphocyte to different doses of β-lg or κ-CN based on PI values after ex vivo stimulation. The increase in β-lg concentrations from 10 to 50 and to 100 μg/mL changed PI from 1.01 ± 0.16, 1.14 ± 0.08, to 1.19 ± 0.08, respectively. The same κ-CN concentrations gave corresponding PI values of 1.31 ± 0.01, 1.92 ± 0.03, and 2.21 ± 0.03. Increasing antigen concentration to 200 μg/mL caused further increase the PI value for β-lg to 1.71 ± 0.01 (*p* < 0.05 vs. 100 μg/mL dose), but decrease for κ-CN to 1.60 ± 0.01 (*p* < 0.05 vs. 100 μg/mL dose). Thus, the optimal concentration was different for β-lg and κ-CN. Previously, we showed similar PI values for naïve mouse lymphocytes stimulated with 200 μg/mL α-CN or κ-CN [18]. Additionally, in this study we did not observe meaningful differences in the PI values for naïve lymphocytes stimulated with β-lg or κ-CN in a dose 200 μg/mL. Finally, despite the effect of the dose on PI, we chose a dose of 200 μg of antigen for in vivo experiments, for oral gavage of mice and further ex vivo lymphocytes stimulation, to maintain a similar research model as before and better results comparison.

### 3.2. Mouse Humoral Response to β-Lg and κ-CN

Specific class G and A immunoglobulins have been implicated in the development of both IgE-mediated and non-IgE-mediated immune responses to food allergens, as well as oral tolerance [22]. Thus, changes in the antigen-specific IgG and IgA levels are useful indicators of the protein immunogenicity [23].

Figure 3 shows dynamic of induction of anti-β-lg and anti-κ-CN IgA and IgG in mice immunized and gavage with β-lg and κ-CN. Serum anti-Ag IgG levels increased throughout the experiment in both groups, reaching the highest titer of 2^16.5±0.85^ for the group κ-CN on day 42 and 2^16.29±0.49^ for the group β-lg on day 49 (Figure 3A). On day 56, serum IgG titers slightly decreased to 2^13.8±0.84^ and 2^15.5±0.58^ for the groups β-lg and κ-CN, respectively. With minor exceptions, no statistical differences were observed between the groups. However, they differed in dynamics of IgA production. Until the 35th day, β-lg-treated mice showed significantly lower serum anti-Ag IgA levels compared to κ-CN-treated mice (*p* < 0.05; Figure 3B). Then, the anti-β-lg IgA levels increased and got EpT 2^7.67±1.52^, 2^8.92±1.92^, 2^7.14±1.52^ on 42nd, 49th, and 63rd day, respectively. κ-Casein presented itself as stronger immunogen, with increased titer of serum anti-κ-CN IgA reaching 2^8.8±1.1^ already on the 21st day post-immunization, 2^9.5±1.41^ on day 28, and 2^10.22±0.83^ on day 35. Similarly, fecal anti-Ag IgA levels were lower for the β-lg group compared to the κ-CN group and amounted to 2^3.75±1.75^ and 2^5.3±1.25^ on day 28, respectively (Figure 3C). Differences between groups were significant (*p* ≤ 0.05) beginning on day 35, with final titers of 2^4.4±0.84^ and 2^6.2±1.3^ for β-lg and κ-CN groups, respectively. Previous studies have demonstrated reduced secretory IgA levels in food allergy mouse models [13,24]. According to Baba et al. [25] low serum IgA levels are good predictors of food hypersensitivity. The humoral response in our study demonstrated relatively higher immunogenicity of κ-CN in comparison with β-lg, i.e., higher serum specific immunoglobulins levels, and a higher level of secretory specific IgA in comparison to β-lg. These findings seem to be in line with the described above PI values, after ex vivo stimulation of naïve lymphocytes (Figure 2). κ-CN dynamically induced lymphocyte proliferation in the concentration range from 10–100 μg (Figure 2B), and the obtained PI values were about 54–83% higher than those obtained for β-lg. The decrease in PI caused by an increase in κ-CN concentration to 200 µg/mL may result from regulatory mechanisms or cell mortality due to excessive antigen stimulation. Some population of T lymphocytes, crucial in maintaining immune resistance and in maintaining tolerance to their own and foreign proteins, are anergic and do not proliferate because of T cell receptor (TCR) stimulation. Independently, these two experiments describe the differences in immunogenic potentials of β-lg and κ-CN. Moreover, these results showed that the mouse’s humoral response to β-lg was lower than to κ-CN. κ-Casein in turn, as we described earlier, gave a lower humoral response than that obtained for α-CN [18]. It may result from differences in IgE binding epitopes on these proteins [26]. Total serum IgE concentration in the presented study was 3 times higher in κ-CN group than in β-lg group (*p* < 0.05; Figure 4). Previously Wróblewska et al. [13] showed that mice sensitized with the α-CN and β-lg mixture had a total IgE level of about 450 ng/mL. However, the action of a mixture of antigens may vary because proteins can trigger various regulatory mechanisms, including a non IgE-mediated gastrointestinal food allergy, known as food protein-induced enterocolitis syndrome (FPIES) [27]. In any case, to sum up the above, based on the humoral response and PI values, it is possible to rank the increasing allergenicity of the cow’s milk allergens in questions in the order β-lg < κ-CN ≤ α-CN.

### 3.3. Milk Antigens Immunoreactivity with β-Lg and κ-CN Activated Splenocytes

The proliferation index reflects the activity of effector lymphocytes and thus the immunogenicity of the protein, and its highest value corresponds to the immunogenic potential of the antigen. To assess if other milk proteins may affect the immune responses activated to β-lg and κ-CN, splenocytes received from β-lg and κ-CN immunized mice were stimulated in culture with α-la, β-lg, α-CN, or κ-CN, and the proliferation index was evaluated using the CFSE method (Figure 5).

The effect of in vivo lymphocyte activation by β-lg and κ-CN was observed ex vivo, using a proliferation assay and milk antigens stimulation. Cell stimulation with the antigens tested increased PI values, and in the β-Ig group, we arranged the antigens with the highest immunogenicity in descending as α-la (PI = 1.74 ± 0.11), β-Ig (PI = 1.66 ± 0.01), α-CN (PI = 1.58 ± 0.08), and κ-CN (PI = 1.46 ± 0.08) (*p* < 0.05 vs. PBS; Figure 5A). Lymphocyte PI measures were similar for α-la, β-Ig, and α-CN. However, stimulation with α-la yielded a difference when compared to κ-CN (*p* < 0.05). The splenocytes of the κ-CN group (Figure 5B) responded stronger than of the β–lg group. The proliferation index showed values in a range from 2.03 ± 0.06 with α-la stimulation to 2.45 ± 0.04 with κ-CN stimulation, and the average PI of κ-CN group lymphocytes was about 0.6 higher (36%) when compared to the β-Ig group PI (*p* < 0.05). So, the ex vivo proliferation experiment confirmed that κ-CN immunization resulted in more allergenic lymphocytes compared to β-Ig immunization. κ-Casein induced a faster antibody response, of greater magnitude (Figure 3) and higher total IgE (Figure 4). Based on the above, we concluded that both immunity and allergenicity of these two proteins differ. Shandilya et al. [28] compared the proliferation of lymphocytes from mice immunized with whole casein or whey protein fraction isolated from cow or buffalo milk, using MTT assay, but their results showed that lymphocytes from the whey protein-sensitized group were more reactive to ex vivo concanavalin A or lipopolysaccharide stimulation. Our experimental model showed that proliferation index determined ex vivo, using diverse milk antigens, could be a good marker for differentiation of protein immunogenicity.

### 3.4. T Cell Profile of Lymphocytes from Inductive and Peripheral Tissue

T cells play a pivotal role in inducing an immune response to antigens. After the first contact with the antigen, naïve CD4^+^ T cells may differentiate into diverse subpopulations, the most important being Th1, Th2, Th17, and induced regulatory cells [28]. Our previous results showed changes in induction of CD4^+^ and CD8^+^ cells after different routes of immunization, suggesting that a percentage of these populations could serve as a marker for screening differences between immunity to different antigens [29]. We found κ-CN as an antigen that effectively induces ex-vivo lymphocyte proliferation. On the other hand, in our previous study ex-vivo stimulated lymphocytes from α-CN-treated mice induced more CD4^+^, CD8^+^, and CD4^+^CD25^+^Foxp3^+^ T cells compared to κ-CN group [18]. Therefore, we evaluated CD8^+^ and CD4^+^ T cells induction in different mouse tissues following β-lg or κ-CN immunization and oral gavage (Figure 6).

Food antigens may induce an immune response via dendritic cells or via M cells, found in PPs and mucosa associated lymphoid tissue, which allow for transport antigens across the epithelial cell layer to the lamina propria, where they are presented to immune cells, and further transport to MLNs, where an immune response is begun. CD4^+^ T cell percent was about 5 times lowered in the MLNs of the κ-CN group compared to the PBS and β-lg groups, and amounted to 9.3 ± 2.5% in comparison to 49.2 ± 9.0 and 48.3 ± 4.2%, respectively (*p* < 0.001) (Figure 6). The mice sensitized with κ-CN also had 2.6 times less CD8^+^ T cells then those sensitized with β-lg, i.e., 7,4 ± 0.65% versus 19.1 ± 0.99%, respectively (*p* < 0.01). In turn, the percentage of CD8^+^ cells in MLNs of the β-lg group slightly increased in comparison to the PBS group (*p* < 0.05). In PPs and SPL, there were no meaningful differences between CD4^+^ and CD8^+^ T cells in the experimental groups, however, with one exception of the CD8^+^ cells in PPs, in both groups there was a decrease in these populations compared to the PBS group (*p* < 0.01). Peripheral immune response in HNLNs showed a higher number of CD4^+^ and CD8^+^ cells in the κ-CN group in comparison to the β-lg group (*p* < 0.05). Nevertheless, there were no differences between the κ-CN group and the control PBS group as regards these populations. A similar tendency was observed in peripheral blood mononuclear cells (PBMCs), although CD4^+^ population in the experimental groups decreased relative to PBS treated mice (*p* < 0.001), and there was no difference in CD8^+^ T cells between β-lg, κ-CN, and PBS receiving mice.

CD4^+^ T cells are precursors for regulatory cells, induced and regulated in various molecular pathways. To show the differentiation of CD4^+^ cells, we checked CD4^+^CD25^+^ and CD4^+^CD25^+^Foxp3^+^ T cells in the experimental group tissues and compare them with those in mice treated with PBS. With minor exception, the induction of CD4^+^CD25^+^ and CD4^+^CD25^+^Foxp3^+^ cells was observed in both experimental groups (*p* < 0.001 vs. PBS group; Figure 5). Nevertheless, the β-lg group had significantly higher levels of CD4^+^CD25^+^ T cells in MLNs, HNLNs, and PBMCs than the κ-CN group (1.9, 11.6, and 2.3 times, respectively; *p* < 0.001). In turn, compared to β-lg group, κ-CN immunization resulted in increased CD4^+^CD25^+^Foxp3^+^ T cells induction in PPs, SPL, HNLNs, and PBMCs (1.2, 2.2, 1.5, and 2.2 times, respectively; *p* < 0.001). The differences between the experimental groups were obviously visible in periphery lymphoid tissue, i.e., in HNLNs and PBMCs, where β-lg induced more CD4^+^CD25^+^, whereas κ-CN induced more CD4^+^CD25^+^Foxp3^+^ regulatory T cells. These results depict cellular responses specific to immunogenic properties of each antigen. Antigen-activated Foxp3^+^ regulatory T cells play a pivotal role in maintaining tolerance to food proteins by modulating the activity of effector T cells [30]. On the other hand, β-lg probably induced more T regulatory type 1 (Tr1) cells, which usually do not express the Foxp3 factor, but have potent suppressive ability through production of immunosuppressive cytokines IL-10 and TGF-β. IL-10-producing Tr1 cells play a central role in the induction of oral as well as systemic tolerance [31]. Sakaguchi et al. [32] were the first to indicate that Tregs regulate CD4^+^CD25^-^ Th cells since their depletion leads to exaggerated CD4^+^ Th cells response, resulting in immunopathology. Tregs have also been shown to regulate other cells of the adaptive immune system such CD8^+^ T cells and B cells [33,34]. We observed a significant decrease in the percentage of CD4^+^ T cells in MLNs of κ-CN group, accompanied by increased humoral response of B lymphocytes and proliferative activity of T lymphocytes (Figure 3, Figure 4, Figure 5 and Figure 6). Lim et al. [33] described direct suppression of B cells by CD4^+^CD25^+^ regulatory T cells. Foxp3^+^ Tregs can migrate to the B cell areas of secondary lymphoid tissue and suppress T cell-dependent B cell Ig response. The differences observed in Tregs suggest different ways of inducing mucosal immune system responses by these two antigens, which was not observed in the previous experimental model when comparing α-CN with κ-CN [18].

The spleen participates in the maintenance of tolerance to antigens and is thought to play a role in oral tolerance [35,36]. Moreover, it is known to act as a tissue repository for resident memory T cell [37]. As such, expansion of Foxp3^+^ Tregs in the spleen could serve as an indicator of the presence of memory subset of antigen-specific regulatory T lymphocytes, which can contribute to the development of systemic tolerance [38].

To confirm the contribution of milk antigens in Tregs induction, splenocytes isolated from the experimental groups were cultured in the presence of α-la, β-lg, α-CN, and κ-CN, thereafter the percentage share of CD4^+^CD25^+^Foxp3^+^ regulatory cells in the whole population of CD4^+^CD25^+^ cells was determined (Figure 7). In both groups, lymphocytes presented the most efficient Foxp3^+^ induction following primary antigen stimulation, i.e., with β-lg for group β-lg (*p* < 0.05; Figure 7A) and κ-CN for group κ-CN (*p* < 0.05; Figure 7B). This confirms the in vivo induced regulatory activity against tested antigens. Savilahti et al. [39] investigated the PBMCs of allergic and non-atopic children and found that only cells of cow’s milk-sensitive individuals induced Foxp3^+^ Treg. Our in vivo study showed that splenocytes from κ-CN group had 2.2 times more Foxp3^+^ regulatory cells than from β-lg group (*p* < 0.001; Figure 6). Similarly, ex vivo stimulation with κ-CN enhanced the Foxp3^+^ cells production in the κ-CN group (*p* < 0.05, Figure 7B), although the effect was smaller than those observed in vivo (22.4 ± 1.98% in comparison to 55.6 ± 1.41% stated in spleen fresh tissue; Figure 6 and Figure 7B). In turn, the cultured splenocytes from β-lg group produced more CD4^+^CD25^+^Foxp3^+^ regulatory cells because of ex vivo stimulation with β-lg, but also with κ-CN (*p* < 0.05), which confirms high immunoreactivity of both proteins, especially κ-CN. The results do not correlate with the described above proliferative activity, where an increase in PI values was found for all milk proteins tested, in both groups (Figure 5). However, this confirms the presence of anergic Tregs, and other lymphocytes induced by the antigens tested, such as B lymphocytes. We observed increased humoral response in both experimental groups, although the effect was higher in the κ-CN-treated mice (Figure 3 and Figure 4). Weiberg et al. [40] showed that transfer OVA sensitized splenocytes to mice followed by oral stimulation with OVA enhanced the migration of the splenic B cells into the gut as well as their switch to IgA+ plasma cells. So, our findings demonstrate the high efficacy of β-lg and κ-CN in expanding CD4^+^ cells in gut-associated lymphoid tissue.

### 3.5. Cytokine Profiles

To characterize induction of T cells with β-lg and κ-CN in more detail, we evaluated the differences in cytokine secretion between the experimental groups (Figure 8). Except for IL-17A, the concentrations of the tested cytokines, including IFN-γ, TNF-α, IL-10, and IL-6 secreted by group β-lg lymphocytes cultured in the presence of β-lg (the primary antigen for the group, black bars marked with rectangle in Figure 8A) significantly increased compared to PBS controls without ex vivo stimulation (*p* < 0.05; Figure 8A). As for the remaining antigens tested, IFN-γ, TNF-α, IL-6, and IL-17A were stimulated by α-la, and the effect observed was almost twice as high as for β-lg (*p* < 0.05). The stimulation β-lg group lymphocytes with κ-CN induced secretion of IL-10 (1132.2 ± 37 pg/mL), in an amount comparable to that observed for the primary antigen (1183.5 ± 113 pg/mL) (*p* < 0.05 vs. other antigens and unstimulated cells). It should be noted that there were no differences between PBS and Con A stimulation, which indicates that IL-10 secretion was generally high in the β-lg group because of previous in vivo exposure. Ex vivo, κ-CN significantly increased IL-17A secretion by lymphocytes of these mice (*p* < 0.05 vs. other antigens and PBS). IL-10 plays multiple roles in mediating the immune response and is particularly involved in Tregs induction and in modulating the balance between allergy and tolerance. Th2 type cells as well as Th1 and CD8^+^ cells secrete IL-10. It inhibits production of inflammatory cytokines such as IL-6 or TNF-α, induces antibody production and Th17 differentiation, and regulates Tregs differentiation [41,42]. Recent evidence suggests the existence of a balance between Tregs and Th17 cells during the development of naïve T cells [43]. The presence of IL-6 and TGF-β promotes the differentiation of T lymphocytes into Th17 cells and IL-17 secretion, whereas without IL-6, T cells differentiate into Tregs. In the β-lg group, elevated concentrations of IL-6 were, respectively, 1952.7 ± 432 pg/mL and 811.8 ± 27 pg/mL after α-la and β-lg stimulation. Meanwhile, IL-17A concentration of 1355.45 ± 210 pg/mL after κ-CN stimulation suggested that β-lg may stimulate CD4^+^ and CD8^+^ cells induction in the IL-6, IL-10, and IL-17 dependent pathways. Dhuban et al. [44] found that sera from allergic patients showed a significant increase of IL-17A production during in vitro stimulation of lymphocytes. Ex vivo β-lg stimulation did not increased IL-17A secretion by group β-lg lymphocytes, but this secretion increased in the presence of other strong milk allergens. It suggests that CD4^+^ lymphocytes from β-lg-sensitized animals have the potential to develop Th2- or Th17-associated phenotypes upon encountering stimulating signals. Peripheral regulatory T cells are less stable than thymus-derived Treg cells and can lose Foxp3 expression and produce cytokines, such as IFN-γ and IL-17A, under inflammatory conditions [45]. The differences in ex vivo cytokine secretion patterns indicate the potential roles of other milk allergens in inducing inflammation in the gut following β-lg immunization, as previously discussed by Wal [46] and Monaci et al. [47].

Splenic lymphocytes from the κ-CN-immunized group showed less dynamic cytokine secretion profiles than those of the β-lg group (Figure 8). Only IL-10 concentration increased due to ex vivo cell stimulation with the primary antigen, i.e., κ-CN (908.22 ± 208 pg/mL vs. to 206 ± 72 pg/mL in PBS group; *p* < 0.05). However, its concentration was smaller than those observed for the β-lg group (Figure 8A). Additionally, stimulation of lymphocytes from the κ-CN-immunized group with α-CN, or β-lg resulted in increase of IL-10 concentration (to 704.3 ± 123, and 547.22 ± 82 pg/mL, respectively), but the effect was smaller than those during κ-CN stimulation (*p* < 0.05). Cells stimulated with α-la or β-lg showed an increase in TNF-α concentrations (to 400.25 ± 80 and 253.5 ± 47 pg/mL, respectively), as compared to the cells cultured with α-CN, κ-CN, or PBS (*p* < 0.05). α-La as the one enhanced IL-6 secretion in this group (*p* < 0.05).

The IL-10 secretion by lymphocytes of both experimental groups was consistent with the strong Treg induction in these groups as described above. However, we observed differences in Tregs levels in peripheral lymphoid tissues (in HNLNs: 4.1-fold and 0.4-fold, and 3.8-fold and 5.6-fold higher for β-lg and κ-CN group, respectively, and CD25^+^ and Foxp3^+^ T cells; *p* < 0.05 in comparison to PBS group; Figure 6), which suggests that the β-lg- and κ-CN-induced immune responses in mice were IL-10-dependent, but the pathways of Tregs induction were different. On average, IL-10 concentration was about 20% higher in lymphocyte cultures from β-lg group compared to κ-CN group (*p* < 0.05), which may result from stronger activation of CD4^+^CD25^+^ cells in this group, probably including Tr1 cells producing this cytokine. β-Lg mouse immunization led to increased production of Th1-associated cytokines, such IFN-γ and TNF-α compared to κ-CN immunization. In turn, immune responses induced by κ-CN were found to be increasingly resistant to the body’s regulatory mechanisms over time, κ-CN-stimulated sensitized cells were less reactive to milk allergens. Despite a high concentration of CD4^+^Foxp3^+^ Tregs, they still produced high amount IgE (Figure 4).

## 4. Summary

In summary, the applied model of in vivo and ex vivo exposure to β-lg and κ-CN facilitated a comprehensive comparison of the immunomodulatory properties of both proteins. The differences in proliferation indexes, and humoral and cellular immunomodulatory responses, not only confirmed the strong immunogenic potential of both β-lg and κ-CN, but also showed differences in induction of regulatory mechanisms. Our results showed that both tested proteins are immunogenic, but κ-CN is a stronger milk allergen than β-lg, considered so far to be one of the strongest milk allergens. The observed differences in Tregs induction, CD4^+^CD25^+^and CD4^+^CD25^+^Foxp3^+^, and in cytokine secretion profiles, make our model a good tool for differentiating the immune potential of food proteins. Furthermore, our ex vivo experiments revealed a risk of interaction of the immune responses to different milk antigens, which may intensify allergy and should be considered in food supplementation. In our opinion, the presented scheme of studies may be particularly useful for comparison of the power of food allergens. Nevertheless, the induction or loss of oral tolerance is modulated by combined mechanisms induced by DCs, Treg cells, B reg cells, and microbiomes, thus further studies are needed to understand and clearly identify the immune response generated by cow’s milk allergens.

## Figures and Tables

**Figure 1 nutrients-13-00349-f001:**
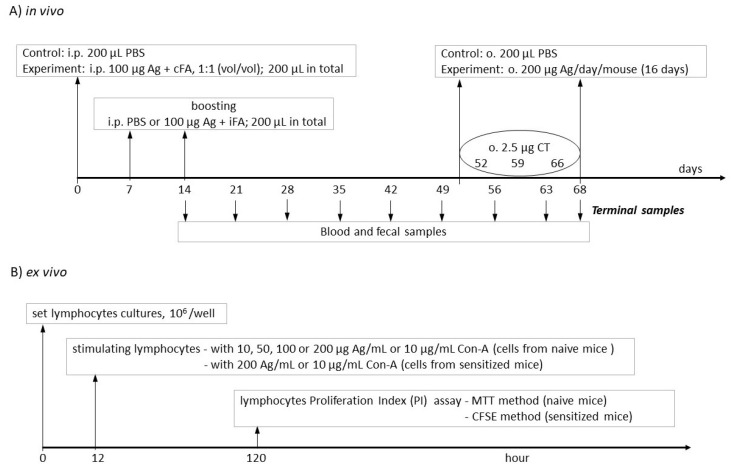
Scheme of experiments design (**A**) in vivo and (**B**) ex vivo. Used abbreviations: Ag—antigen; cFA—complete Freund adjuvant; iFA—incomplete Freund adjuvant; CT—cholera toxin.

**Figure 2 nutrients-13-00349-f002:**
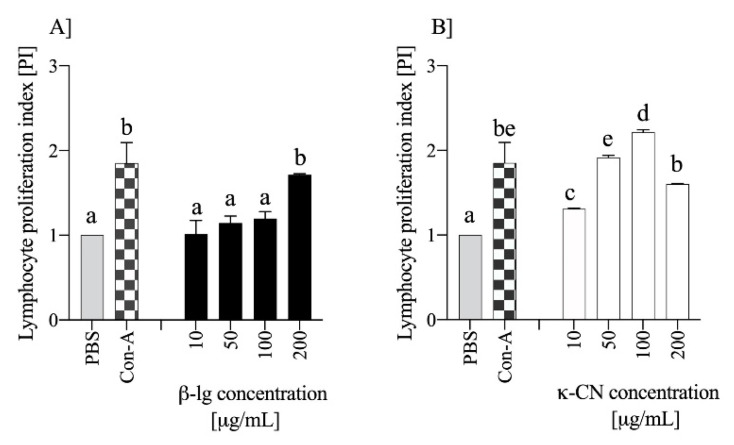
The effect of antigen dose on proliferation index (PI) of naïve splenocytes stimulated with different doses of β-lg (**A**) or κ-CN (**B**) after 120 h culture. Cells were obtained from non-immunized mice. PI was determined with MTT method and calculated as the ratio of mean absorbance of antigen stimulated cells to mean absorbance of unstimulated cells. Concanavalin A (Con-A) was used as a positive control. The data are presented as the means ± SD. Statistical analysis was performed with one-way ANOVA followed by post hoc Tukey test. Different letters present statistical differences between means at *p* ≤ 0.05.

**Figure 3 nutrients-13-00349-f003:**
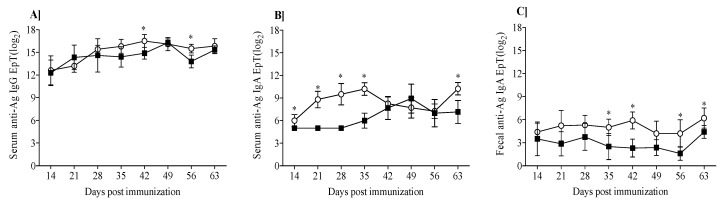
Specific antibody titers induced in mice after immunization with β-lg (black squares) and κ-CN (white circles): (**A**) Serum anti-Ag IgG, (**B**) serum anti-Ag IgA, (**C**) fecal anti-Ag IgA. Each data point corresponds to the mean of the group ± SD. Statistical analysis was performed by *t* test. Asterisks indicate differences between the two groups at the same data point at *p* ≤ 0.05.

**Figure 4 nutrients-13-00349-f004:**
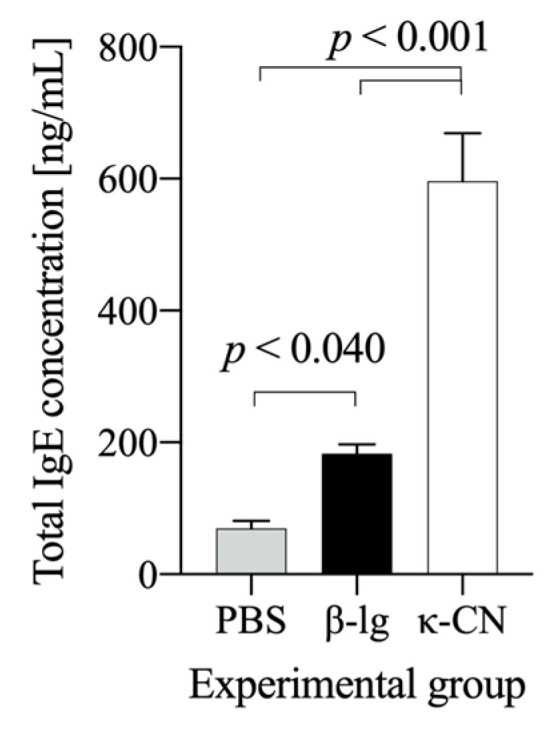
Total serum IgE concentration in mice immunized with β-lg (black bar) and κ-CN (white bar). The data are expressed as means ± SD. Statistical analysis was performed with one-way ANOVA follow by Tukey’s post hoc test.

**Figure 5 nutrients-13-00349-f005:**
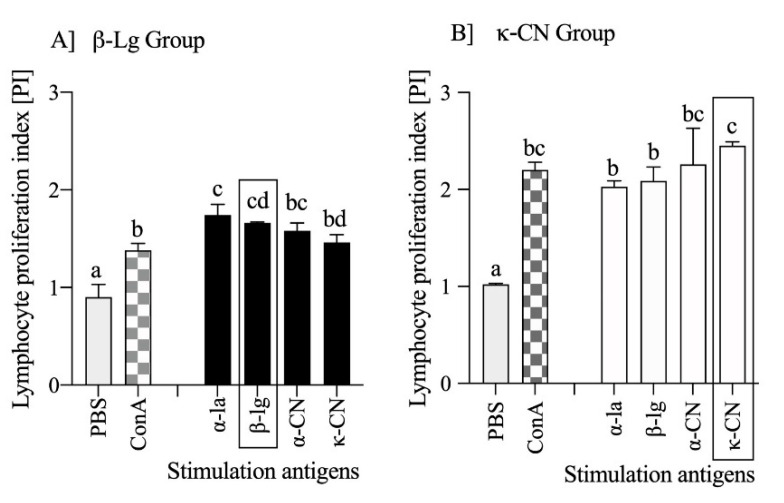
The effect of major milk antigens (dose 200 μg/mL) on splenocytes sensitized in vivo to β-lg (**A**) or κ-CN (**B**). Proliferation was assessed via flow cytometry using the CFSE method. The data are expressed as means (*n* = 5) ± SD. Statistical analysis was performed with one-way ANOVA follow by Tukey’s post-hoc test. Different letters present statistical differences between means at *p* ≤ 0.05. A rectangle around the bar assigns primary antigen used for cells stimulation.

**Figure 6 nutrients-13-00349-f006:**
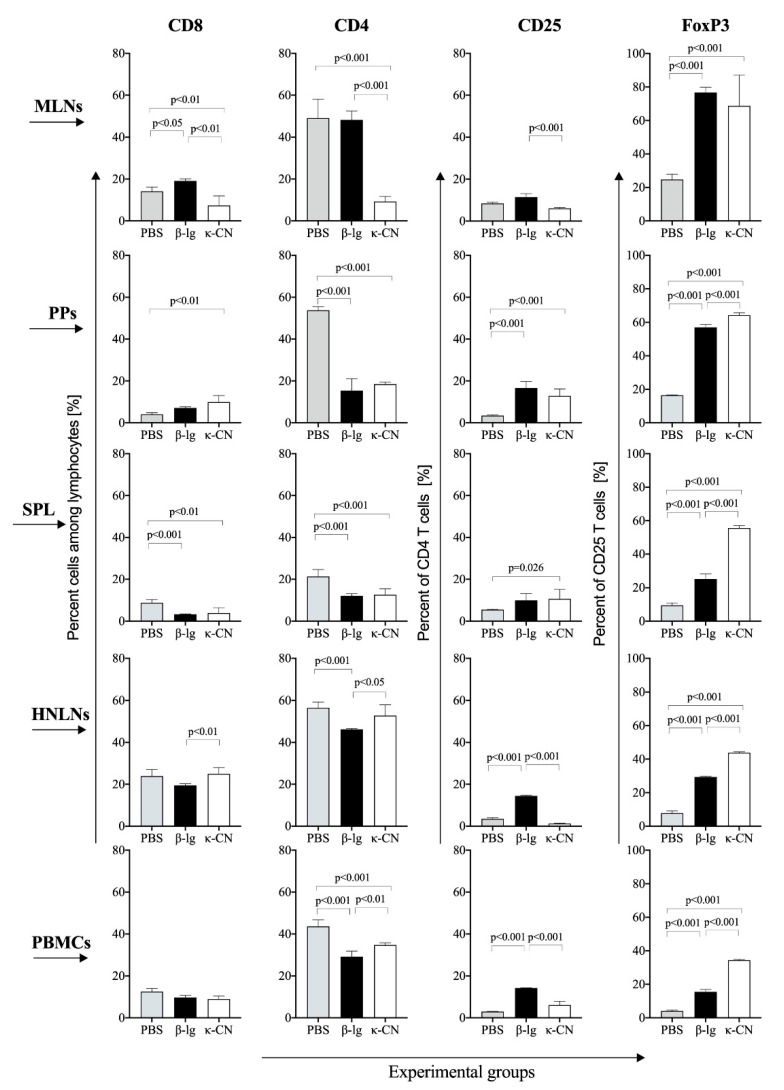
Distribution of CD8^+^, CD4^+^, CD4^+^CD25^+^, and CD4^+^CD25^+^Foxp3^+^ T cells in mesenteric lymph nodes (MLNs), Peyer’s Patches (PPs), spleen (SPL), head and neck lymph nodes (HNLNs), and peripheral blood mononuclear cells (PBMCs) of mice immunized with β-lg (black bars) or κ-CN (white bars) and control mice treated with PBS only (gray bars). The data are expressed as the means ± SD. Statistical analysis was performed with one-way ANOVA follow by Tukey’s post-hoc test.

**Figure 7 nutrients-13-00349-f007:**
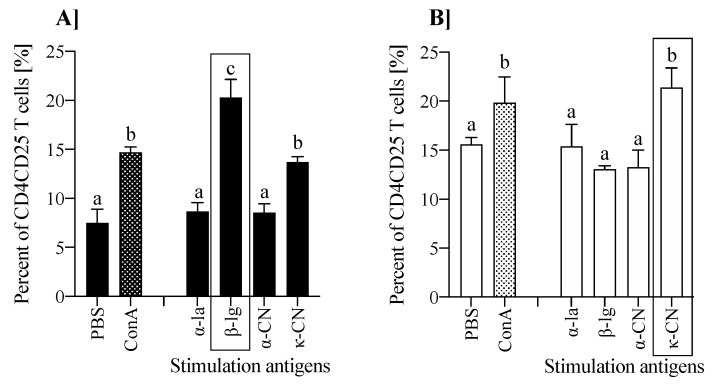
The effect of ex vivo stimulation of splenocytes isolated from mice immunized with β-lg (**A**) or κ-CN (**B**) with different milk antigens (at a dose 200 µg/mL) on the induction of CD4^+^CD25^+^Foxp3^+^ cells. Data are expressed as a mean ± SD. Statistical analysis was performed with one-way ANOVA follow by Tukey’s post-hoc test. Different letters present statistical differences between means at *p* ≤ 0.05. A rectangle around the bar assigns primary antigen used for cells stimulation.

**Figure 8 nutrients-13-00349-f008:**
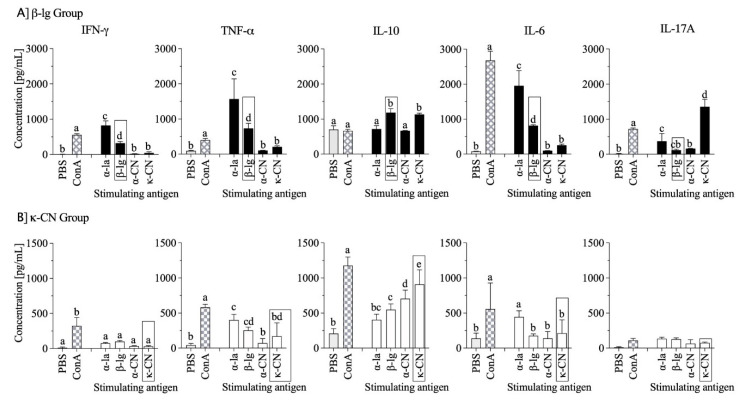
Cytokines secreted during splenocyte culture after ex vivo stimulation with different milk antigens (dose 200 µg/mL). Lymphocytes were isolated from mice immunized with β-lg (**A**) or κ-CN (**B**). Data are expressed as mean ± SD. Statistical analysis were performed with one-way ANOVA follow by Tukey’s post-hoc test. Means with different letters differ at *p* < 0.05.

## Data Availability

Not applicable.

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
