# Peer review of "Differences in Regulatory Mechanisms Induced by β-Lactoglobulin and κ-Casein in Cow’s Milk Allergy Mouse Model–In Vivo and Ex Vivo Studies"

_nutrients, 2021, doi:10.3390/nu13020349_

Round 1

Reviewer 1 Report

The authors aimed to estimate the immunomodulatory effects of β-lactoglobulin (β-lg) in comparison to those elicited by κ-casein (κ-CN), in vivo and ex vivo, using naïve splenocytes and mouse sensitization model.

Comments:

1.Line 101, 2.2. Mice and treatment groups

Six-weeks-old female BALB/ccmdb mice were purchased….

Why not use both sexes?

Inclusion of both sexes in animal research studies should drive important discoveries in both basic and clinically relevant research (Proc Natl Acad Sci U S A. 2015 Apr 28;112(17):5257-8. PMID: 25902532).

2.Figure 5. Line 328, Statistical analysis was performed with “one-way ANOWA” follow by Tukey’s post-hoc test.

It should be a typing error, …with “one-way ANOVA”.

Author Response

Dear Reviewer,

thank you for your work. Any comments will translate into a better final job, so thank you again.

Corrected and newly added text in the manuscript is highlighted using the “Track Changes” function.

Regards

Ewa Wasilewska – corresponding author

Response to Reviewer 1 comments

Comments:

1.Line 101, 2.2. Mice and treatment groups

Six-weeks-old female BALB/ccmdb mice were purchased….

Why not use both sexes?

Inclusion of both sexes in animal research studies should drive important discoveries in both basic and clinically relevant research (Proc Natl Acad Sci U S A. 2015 Apr 28;112(17):5257-8. PMID: 25902532).

AU: Mouse models are indispensable to better understand food allergies and establish novel treatment options. Using them we may investigate allergic pathology, compare allergenic potential of dietary ingredients and test the potency and safety of novel options and vaccines. Currently, food allergies are treated with a combination of allergen avoidance and symptomatic treatment. Despite of many mouse models of food allergy applied for scientific studies, none is perfect/fully sufficient. There is no naturally occurring allergic mouse. The induction of sensitization and allergic reactions in mice will always be an artificial process. Furthermore, apart from similarities in the immunological mechanisms, certain differences exist that should be considered when translating results from mouse to human. Many factors influence sensitization success in mice, such route of application, dose, number and frequency of application, applied adjuvant, and finally environmental and host factors, such diet, microbiome, drugs, mouse strain, sex, age, or genetic differences. However, one thing is common and certain about mouse models, they should reproducibly portray the immunological parameters of food allergy. As regards allergic sensitization, male mice exhibit greater variability than female mice, the latter are more susceptible to the development of allergic inflammation (our unpublished data; Melger et al., 2005; Schaefer et al., 2020). Female mice are used in laboratories across the world to study food allergy.

2.Figure 5. Line 328, Statistical analysis was performed with “one-way ANOWA” follow by Tukey’s post-hoc test.

It should be a typing error, …with “one-way ANOVA”.

AU: Thank you. Corrected

Reviewer 2 Report

The present work provides information about an in vivo and an ex vivo murine model to study the regulatory mechanisms induced by cow’s milk allergy, and the authors propose such model as suitable for further study of other food allergens.

The structure chosen by the authors, bringing together Results and Discussion makes it easier for the reader to contextualize their results.

The global message of the manuscript is clear and brings solid pieces of evidence and data. Nevertheless, the authors do not mention the limitations of the study.

Minor comments

  1. A line or two about the strengths and limitations of the study in the summary section are required.
  2. In lines 55 and 61 CMA is mentioned although it is not explained what stands for.
  3. There are some misspellings in lines 117 (routE), 118 (sujectED), 269 (Lavage), 328 (ANOVA)
  4. In line 239, TO particle is missing before “be used”
  5. In line 483, “The others“ should be replaced for a better referential expression.

Author Response

Dear Reviewer,

thank you for your work. Any comments will translate into a better final job, so thank you again.

Corrected and newly added text in the manuscript is highlighted using the “Track Changes” function.

Regards

Ewa Wasilewska – corresponding author

Response to Reviewer 1 comments

Minor comments

  1. A line or two about the strengths and limitations of the study in the summary section are required.

AU: Corrected. See line 539-542.

  1. In lines 55 and 61 CMA is mentioned although it is not explained what stands for.

AU: Thank you. Corrected

  1. There are some misspellings in lines 117 (routE), 118 (sujectED), 269 (Lavage), 328 (ANOVA)

AU: Thank you. Corrected

  1. In line 239, TO particle is missing before “be used”

AU: Thank you. Corrected

  1. In line 483, “The others“ should be replaced for a better referential expression.

AU: Thank you. Corrected

Reviewer 3 Report

Congratulations on an interesting job. If I understand correctly, the increase in the concentration of the antigen proportionally increases the proliferative index of the lymphocytes, however regardless of this it is the casein that produces a greater IgE mediated response.

Finally was the difference after heat exposure of the proteins also evalueted ? Thanks for the relapy

Author Response

Dear Reviewer,

thank you for your work. Any comments will translate into a better final job, so thank you again.

Corrected and newly added text in the manuscript is highlighted using the “Track Changes” function.

Regards

Ewa Wasilewska – corresponding author

Response to Reviewer 3 comments

Congratulations on an interesting job. If I understand correctly, the increase in the concentration of the antigen proportionally increases the proliferative index of the lymphocytes, however regardless of this it is the casein that produces a greater IgE mediated response.

AU: Proliferation index due to stimulation of naïve lymphocytes with antigen may be used for initial screening of protein immunogenicity. Following antigen co-stimulation, activated T cells proliferate rapidly, such that the number of cells from a single clone increases exponentially. The increase in the antigen concentration enhance T cells activation, but  serial dilutions of antigen should be tested, because, as was seen with casein, too high concentration of antigen can inhibit cell growth in vitro. Also, in vivo, although allergens behaves differentially in general, lower allergen amounts were repeatedly shown to have a higher sensitizing potential than corresponding higher allergen doses. T cell activation coincides with changes in cellular metabolism that must be coordinated with instructive signals from cytokine an costimulatory receptors to generate an immune response. We observed higher proliferation indexes for κ-CN, at lower antigen doses, compared to β-lg, which corresponded to higher IgE secretion by κ-CN sensitized mice.

Finally was the difference after heat exposure of the proteins also evalueted ? Thanks for the relapy

AU: Heat treatment is a particular process in the food industry and its influence on protein properties is investigated across the years. Developing new technologies, changes in the food matrix make ‘heating processes’ important to evaluate by scientists. Heating alters the structure of high-order protein, which effects their physical and chemical properties and biological activity, including immunoreactivity or immunogenic potential. Our previous experiments showed in models that even termization or pasteurization altered the tertiary structure of milk proteins, which increased their immunoreactivity under certain conditions (Mierzejewska et al., 2002, Milchwissenschaft-Milk Sci. Int. 2002, 57, 9–13; and 2003, Acta Aliment. 2003, 32, 237–246,  https://doi.org/10.1556/AAlim.32.2003.3.3). We are no longer conducting this research. However, current laboratory techniques allow to study this topic deeply at the cellular and molecular level (cytometry, mass spectrum, etc.).